# Fully Open-Access Passive Dry Electrodes BIOADC: Open-Electroencephalography (EEG) Re-Invented

**DOI:** 10.3390/s19040772

**Published:** 2019-02-13

**Authors:** Gaetano D. Gargiulo, Paolo Bifulco, Mario Cesarelli, Alistair McEwan, Armin Nikpour, Craig Jin, Upul Gunawardana, Neethu Sreenivasan, Andrew Wabnitz, Tara J. Hamilton

**Affiliations:** 1The MARCS Institute, Western Sydney University, Milperra, NSW 2214, Australia; andywabnitz@gmail.com; 2School of Computing, Engineering and Mathematics, Western Sydney University, Penrith, NSW 2747, Australia; u.gunawardana@westernsydney.edu.au (U.G.); N.Sreenivasan@westernsydney.edu.au (N.S.); 3Department of Electrical Engineering and Information Technologies, University “Federico II” of Naples, 80121 Naples, Italy; pabifulc@unina.it (P.B.); cesarell@unina.it (M.C.); 4School of Electrical and Information Engineering, The University of Sydney, Camperdown, NSW 2006, Australia; alistair.mcewan@sydney.edu.au (A.M.); craig.Jin@sydney.edu.au (C.J.); 5Sydney Medical School, Central, Royal Prince Alfred Hospital, Camperdown, NSW 2006, Australia; armin@sydneyneurology.com.au; 6School of Engineering, Macquarie University, Ryde, NSW 2113, Australia; tara.hamilton@mq.edu.au

**Keywords:** dry electrodes, brain computer interface, electroencephalography, BCI, EEG, BCI2000

## Abstract

The Open-electroencephalography (EEG) framework is a popular platform to enable EEG measurements and general purposes Brain Computer Interface experimentations. However, the current platform is limited by the number of available channels and electrode compatibility. In this paper we present a fully configurable platform with up to 32 EEG channels and compatibility with virtually any kind of passive electrodes including textile, rubber and contactless electrodes. Together with the full hardware details, results and performance on a single volunteer participant (limited to alpha wave elicitation), we present the brain computer interface (BCI)2000 EEG source driver together with source code as well as the compiled (.exe). In addition, all the necessary device firmware, gerbers and bill of materials for the full reproducibility of the presented hardware is included. Furthermore, the end user can vary the dry-electrode reference circuitry, circuit bandwidth as well as sample rate to adapt the device to other generalized biopotential measurements. Although, not implemented in the tested prototype, the Biomedical Analogue to Digital Converter BIOADC naturally supports SPI communication for an additional 32 channels including the gain controller. In the appendix we describe the necessary modification to the presented hardware to enable this function.

## 1. Introduction

Electroencephalography (EEG) recordings are increasingly being used in numerous applications, ranging from typical clinical applications, e.g., the diagnosis and treatment of brain function anomalies such as seizures, to experimental applications like brain computer interfaces (BCIs) and neurofeedback, which aim to either augment the effective pathways of the human brain by adding non-standard output channels via a computer program, or to increase the performances of a wearer, i.e., meditation feedback [1,2,3,4,5,6,7,8,9,10,11]. For these reasons, BCI/neurofeedback systems are becoming increasingly popular. There are many open-access/source BCI systems that allow those with neurological impairment to have effective communication, as well as systems to aid able-bodied subjects who want to explore the benefits of direct brain to computer communication [12,13,14,15,16].

A substantial difficulty with recording EEGs is that the signal itself is of a very small amplitude, usually in the order of a few tens of micro-Volts (smaller for evoked potentials used for some BCIs) and are easily corrupted by noise. This necessitates careful scalp/skin preparation, the use of conductive gel/paste, and ultimately, glues, to hold the electrodes firm to the skin and avoid the skidding of electrodes [3,17,18,19]. The use of gel/paste and glue can lead to additional problems, such as conductive gel desiccation, and a loss of adhesion leading to an increase in the contact impedance between the electrodes and scalp, which in turn may lead to a large reduction in the signal-to-noise ratio. Short circuits can also occur between neighboring electrodes due to sweat or smearing of the conductive gel [10,19].

Many of these additional problems can be minimized by using dry electrode systems, and some recent studies have shown working prototypes also suitable for EEG [1,4,8,20,21,22,23,24,25,26,27,28,29,30]. Of course, the use of fully “loose” dry electrodes is not immune from artifacts. In previous studies we demonstrated a reduction in the effect of momentary loss of scalp contact due to movements and/or skin stretching by scalp muscle artifacts (i.g., blinking) by developing a lightweight custom-made dry electrode that, by being flat and flexible, was suitable for direct gluing onto the scalp using standard collodion glue cured with a brief oxygen application (readily available in the hospital prep room used for our clinical study) [4,10,19]. 

Recently, gold plated or Ag/AgCl hair penetrative electrodes that look like small hair-combs, with rounded spikes capable of effectively penetrating hair to avoid hair matting under the electrode, have become commercially available (i.g. the gTEC Sahara electrodes http://www.gtec.at/Products/Electrodes-and-Sensors/g.SAHARA-Specs-Features and Ag/AgCl dry electrodes https://shop.openbci.com/); when these electrodes are paired with a suitable analogue front-end, they can measure EEG without the use of conductive gel/paste or glue. To hold the electrode in the designed places, either specialized EEG-caps or standard rubber tube “head nets” can be employed; connection to the electronic front-end is usually made with alligator or pop-fastener clips.

To allow compatibility with both standard wet electrodes as well as commercially available and “home-brew” dry electrodes, such as ones made of spring-loaded pins, jacks, or other “Do It Yourself” solutions that are quite popular among OpenEEG users (http://openeeg.sourceforge.net/doc/hw/electrodes/passive/), we designed an 8-channel module (users can daisy-chain modules in multiples of eight channels) based upon our early passive dry electrode technology [4,22,31,32,33,34,35]. The front-end that we designed is immune to contact impedance imbalance, and in previous trials on animals it has been able to detect fetal electrocardiographic signals in pregnant dairy and beef cattle without the need to prepare/shave the cow using hair penetrative electrodes similar to the ones that now are commercially available for EEG purposes [36]. 

Like our previous designs, the front-end exploits the peculiarity of the INA116 instrumentation amplifier [37] to enable reference electrode replication without additional hardware (see methods section) [4,20,21,22,23,31,32,33,36,38]. Although we previously implemented a modified voltage supply bootstrap to make the driven right leg electrode and/or body grounding unnecessary [39], in this design we also implemented a driven right leg connection as an optional aid for further improvement of the signal to noise ratio that could be of use in difficult EEG/BCI recordings that make use of evoked responses known to be among the faintest EEG signals, requiring signal averaging to be clearly measured [9,40,41,42,43,44,45,46,47,48,49,50].

In order to process the acquired EEG signals, the analogue signals need to be digitized and sent to a computer that is the central processing unit of a BCI. For this purpose, we have selected BCI2000 (www.bci2000.org), which is probably the most well-known, general purpose rapid prototyping BCI software [14]. In order to use BCI2000, a specialized bridge software is required to interface the digital converter of choice to the core of the software. Several commercial and popular hardware solutions (including the Open-EEG hardware http://openeeg.sourceforge.net/), are already included in BCI2000; in order to use our BIOADC we needed to write our own bridge software using the supplied code template [14,51,52]. The BIOADC is an open source, 32-channel analogue to digital converter (ADC) that supports programmable gain control for all the channels. In our original implementation, the system was intended to be stand-alone; hence, it was paired with a supporting ARM processor to record the data on an SD card. However, the device can be also connected (with galvanically isolated connection) to a standard USB port and hence to a standard computer [51]. To allow full reproduction of the hardware, all the production files and bill of materials for both boards, BIOADC, and the analogue electrodes’ front-end are included together with the firmware and the necessary BCI2000 device driver as both source code and compiled code.

With this work, taking all the best features of our previous analogue front-end designs, we present a single module with ultra-high input impedance, suitable for virtually any dry electrode, that allows for simultaneous use of differential inputs, as well as unipolar inputs and user programmable gain when combined with the BIOADC, which is also fully compatible with BCI2000. Therefore, in this paper, we will briefly summarize the hardware design for the BIOADC limited to the board used—the original BIOADC used two (see Section 2.2)—as well as for the dry electrodes’ front-end that is a new design (see Section 2.1); we then present the validation bench tests employing suitable signal generators, as well as some EEG tests limited to alpha-wave elicitation that demonstrates the suitability of our proposed hardware for EEG measurements [49]. 

## 2. Methods

In this section we present the electronic hardware for both the BIOADC and the analogue front-end; we then explain, in detail, the manufacturing of the electrode cap used for our experiments together with all the necessary bench tests and alpha-wave elicitation tests. In appendix we also present an improvement to the proposed hardware to enable digital gain control as well as a comparison of our hardware with some commercially available devices.

### 2.1. Analogue Front-End

The analogue front-end is designed around the INA116 instrumentation amplifier by TI [37]. The schematic for a single channel is depicted in Figure 1, where the INA116 is marked by the designator U1. As can be observed from the schematic, the front-end is fully differential. To enable reference to a single location on the head (e.g., linked mastoid, nose tip, or any arbitrary head electrode), we exploit the embedded “active-guard” [38,51,52] feature present inside the INA116. In other words, one of the channels is configured as fully differential; then the signal from the electrode connected to the inverting input once buffered by the internal guard amplifier is fed to all the other channels’ inverting inputs. Although this design inherits several features from our previous designs [37,52], this implementation is new and purposely designed for modular EEG/BCI and to suit the BIOADC.

In order to configure the selected channel to share the same reference signal, the jumper labeled “In-D” (see Figure 1) must be closed, R3 populated, and the jumper labeled “In-S” open. When In-D is closed, the signal at the output of the inverting input’s guard buffer can be used as a reference for all the other channels. When In-S is closed, the inverting input of that selected channel is connected to the reference signal, i.e., the one from an In-S of another channel. To prevent the general reference electrode from being accounted multiple times in the voltage supply bootstrap and/or in the Driven Right Leg (DRL), the resistor R3 should be populated only for the channel that is assumed as reference. The reference replication technique is depicted in Figure 2, limited to two channels. 

The weighted sum of all the connected electrodes is used to directly drive the battery common node, thereby avoiding the connection of other electrodes to the head/body. To achieve this configuration (modified voltage supply bootstrap that we successfully employed in other biopotential circuits [10,11,19,31,32,33,34,35,36,38,40,51,52]), the selector jumper labeled “RLD-S” (see Figure 3) must be switched to short pins one and two; then, the “RLD” header should be shorted using a jumper. In this configuration, a direct body ground is possible, but an in-line resistor of 100 kΩ should be used between the electrode and the return ground to protect the circuitry from electrostatic discharge. Direct body grounding can be safely employed only when the circuit is battery powered and connected to a galvanically insulated analogue to digital converter like the BIOADC (see next section) [2,51,52]. The same weighted sum of all the connected electrodes can be employed as base signal for a proper DRL. For this reason, we have provided a DRL amplifier with a split resistor design (part of the limiting resistor is inserted into the amplifier feedback loop), designed according to the specs present in reference [2]. Please note that the value of resistance needs to be adjusted according to the user needs (e.g., power supply max patient current, etc.). The use of the DRL is well known to reduce the power-line noise capture via common mode coupling, e.g., irradiation of leads, induction, etc. The effective reduction of power-line noise capture is a function of the type of noise capture and body conditions; therefore, the user should verify that the calculated values of resistance are suitable for the application [1,2,3,6].

For this design, all the operational amplifiers are OPA244 [53]. This particular operational amplifier was selected because it is low power and thus suitable for battery powered devices. In addition, it is immune to “phase inversion” and can thus be used in a non-inverting configuration. The necessary band-pass characteristic is ensured by a passive hi-pass cell (see C5 and R5 in Figure 1), whereas, the resistor R5 also ensures necessary biasing for the operational amplifier U2. U2 is mounted as a non-inverting, first order low pass filter with a DC gain of 100 V/V; the low-pass characteristic is given by R7 and C7, thus fixing the theoretical low-pass cut-off frequency below 50 Hz. The final gain of the EEG front-end is fixed to 1000 V/V using a multi-turn potentiometer connected as an INA116 gain resistor (see bench-test section). The necessary anti-aliasing filter is provided by the BIOADC (see next section).

As it is manufactured, the 8-channel module provides one channel (CH0 see Figure 3) as a reference channel and a header connector (In-AUX) that allows the connection of a reference signal from another module. The reference signal from another module is connected to the In-S of the other modules. Connection of the other eight electrodes to the averaging bootstrap is naturally made via the circuit ground, and the user can use any RLD circuitry present on any module.

### 2.2. BIOADC

At the core of BIOADC there is a supporting processor (PIC18F46J50). This PIC, aside from being low-power, has been selected because it allows for direct USB connection to a master unit (PC in this case, ARM processor for stand-alone applications [51]). This PIC microprocessor also includes a dual SPI/I2C interface which can be used to control, as in our case, four SPI ADCs and a number of SPI digital potentiometers or directly programmable gain amplifiers (PGAs). To enable this function, up to 32 chip-select lines (CS) are supplied together with all the necessary digital communication lines to implement full SPI control via four connectors populated on the right-hand side of the board (see Figure 4). Full galvanic insulation of the BIOADC board is achieved via a USB2.0 isolated HUB (ADuM4160); Power for the BIOADC board is directly drawn from the USB via galvanically insulated DC-DC converters (ADuM5000W) [51], see BIOADC bill of materials for full specifications and detailed part numbers.

Necessary anti-aliasing low-pass filtering is embedded into the BIOADC for a sampling frequency of 1000 Hz. This is simply made as per specification of the selected ADC (LTC1859) using a carefully selected ceramic capacitor in parallel with each ADC input. These capacitors are populated on the BIOADC board. Please note that the capacitors listed in the BIOADC bill of materials (see Appendix A) should be taken as guidelines and their values adjusted for the selected maximum sample rate. 

The PC driver originally supplied with BIOADC works with any Windows 7/8 PC and has been used to develop the necessary BCI2000 EEG source. The compiled driver, as well as the source code and compiled version of the BCI2000 EEG source, is available for download as additional material to this paper.

### 2.3. Dry Electrodes

For this project, we have selected the low-cost Ag/AgCl hair penetrative electrodes available for sale from https://shop.openbci.com/ (depicted in Figure 5). To take full advantage of the active shield, we decided to connect the electrodes using thin shielded cable normally used for portable headphones and used a standard RCA shielded connector to enable easy assembly of the prototype box. For the assembly we employed sections of cable of the same length. Different lengths of cable (different capacitance), however, are not a problem for the INA116, as it is designed for high impedance sensors (including capacitive). Most importantly, the parasitic capacitance of the dry electrodes, because of the lack of smoothing conductive fluids, is known to be quite high and random [1,3,10], which would dominate the small additional parasitic capacitance of connecting cables.

To place the electrodes on the head, we simply re-used the ventilation holes normally present on a standard baseball cap. To enable rough standardized position of our EEG cap on the head, we used the top button of the cap as a reference to be placed in correspondence with the standard EEG electrode position Cz. The button was removed to enable visual localization of the Cz position, which should be marked on the head as per standard EEG preparation. Of course, the position of electrodes is not precise. However, for this paper we are not performing any BCI task and only recording alpha waves to show the EEG recording capabilities of our device. To perform proper BCI, electrode locations should be carefully identified and/or a suitable EEG cap should be used.

## 3. Results

As mentioned, before testing on a volunteer subject, each of the eight channels was calibrated to 1000 V/V using a differential signal generator (Function Generator Model 220 by MEDICAL Instruments, Columbus, Ohio), capable of generating precise sine waves of 1 mVpp at the EEG frequency range. For calibration the circuit reference ground was tied to the signal generator ground using a flying lead connected to the shield of the DRL electrode connection. Noise level (input shorted) was measured and confirmed as less than 3 µV_pp_ in the bandwidth 0.1 to 10 Hz. This value is in line with INA116 and OPA244 specifications. The full summary of electrical characteristics appears in Table 1.

Without performing any head preparation, the makeshift EEG cap described in the previous section was pushed onto the volunteer’s head. The recording was intentionally performed in the late afternoon (at least seven hours after the last showering), the hair was roughly two centimeters in length, and no hairdressing products were used since early morning. As it is possible to observe from Figure 6, two electrodes (represented with bold traces) were in intermittent skin contact. They correspond to the lateral front most aerating holes and due to the shape of the particular cap used, they were in contact with the scalp only if a gentle pressure was applied. As it is possible to observe the jaw artifact (intentional), using the scalp muscle was enough to shift one of the electrodes into contact (see electrode labeled 3 in Figure 6). As we are not performing any real BCI/EEG tests in this paper, during the alpha wave test we disregarded these electrodes.

Like any biopotential, the EEG is affected by power-line noise; all the data presented are pre-processed to remove the power-line (50 Hz) and its harmonics up to the Nyquist frequency using Infinite Impulsive Response (IIR) notch filters. Baseline drift and further unwanted high frequencies are removed using an IIR band-pass filter operating in the 0.6–35 [Hz] range. All filters have an order of fifty and are run in a non-causal manner to zero any phase delay.

The alpha wave test is shown in Figure 7 for both DRL (panel a) and direct grounding (panel b). In both cases, the reference electrode was at the left ear lobe while the grounding/DRL electrode was at the right ear lobe. Although alpha waves were visible over all the head, we selected an electrode placed in the occipital region (see the red trace in Figure 8) corresponding to a position very close to O1 of the 10–20 system. As it is possible to see in Figure 8, this setup exhibited clear alpha wave bursts; notably, the electrode corresponding to channel 2 was very close to the position O2, and like the red marked channel, also presented prominent alpha bursts. In both cases, we measured the power increase in the alpha bandwidth as a ratio of the inverse of the max alpha power divided by the power of the EEG at the same frequency. For the data depicted in Figure 7a, the ratio scores 1.30 [10log_10_(µV^2^/Hz)/10log_10_(µV^2^/Hz)]; for the data in Figure 7b, the ratio is 1.31 [10log_10_(µV^2^/Hz)/10log_10_(µV^2^/Hz)]. Ratio calculations have been rounded to two decimal places. As expected, the max power amplitude of the alpha wave activity is common and, in this case, is 11 Hz. The small discrepancy between the power increases is justified by the different levels of baseline activity that are expected when the subject has his or her eyes open [1,2,3].

To fully evaluate the effect of the DRL, we also assessed the noise capture for the EEG data, recorded directly, comparing several 20 s sections of raw data (before filtering). Generally, limited by the environmental test conditions (air-conditioned electronic lab with fluorescent lighting equipment), we found that the use of the DRL reduced the power-line noise capture by at least 10 dB (measured as 10log_10_(µV^2^/Hz)). Figure 9 depicts an example of such comparisons. As it is possible to compare the height of the 50 Hz components, the DRL, as expected, is shown to reduce the noise capture.

## 4. Conclusions and Discussion

We have presented a fully open access multi-channel (up to 32 channels) alternative to the popular Open-EEG. This version can use virtually any kind of passive electrodes. The proposed embodiment is for EEG measurements; however, it can be modified via jumpers on the PCB to use fully differential channels, thereby allowing recording of other biopotentials. Mixing other biopotentials with EEG enables implementation of human-computer interfaces that still use the popular BCI2000 software (providing that the user customizes the application and signal processing modules). Although limited, our EEG signal testing clearly shows that our device is capable of recording brain-waves, even using a make-shift EEG cap built around a standard one-size fits-all baseball cap. While the use of makeshift EEG caps is attractive (as it could enhance the wear-ability of the device thus promoting mobile BCIs), the final user should consider the use of a properly built EEG cap that allows precise electrode placement, which will enable reproducibility of BCI tasks that are highly dependent from the specific brain area, e.g., P300 BCIs.

Although precise micro-volts measurements of EEG potentials are normally used only in clinical practice, being able to increase/reduce the signal gain according to the experimental condition is a desirable feature. For this reason, the proposed hardware can be further modified to enable programmable gain amplifier (PGA) functions and software fine gain tuning (see Appendix B).

As with every design, there are limits and drawbacks. The major limitation of this design is its size. One may note that many of the proposed functions, such as multiple channels, PGA functionality, and direct analogue to digital conversions, are presently implemented in integrated biopotential front-ends like the ADS1299 by TI (http://www.ti.com/product/ADS1299). However, fully integrated solutions, while they can be cost effective and size effective, may not be compatible with passive dry electrodes, and may come in packages (e.g., Ball Grid Array BGA) that are prohibitive for labs where electronic support is not available. For Instance, ADS1299 bias current is 300 pA; this value is due to a small input impedance that may not be suitable for dry passive electrodes. In addition, the proposed analogue front-end is not entirely dedicated to EEG electrodes, allowing unipolar channels to be mixed with differential channel, and enabling the full human interface paradigm within the popular BCI2000 software. In Appendix C a comparison of our proposed hardware with some commercially available is detailed together with an example of BCI application.

## Figures and Tables

**Figure 1 sensors-19-00772-f001:**
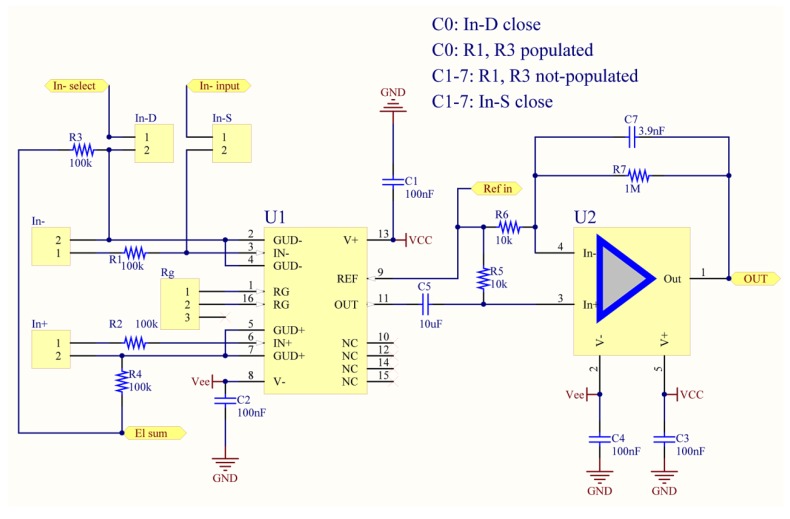
Analogue Front-end schematic (screen capture from Altium Designer).

**Figure 2 sensors-19-00772-f002:**
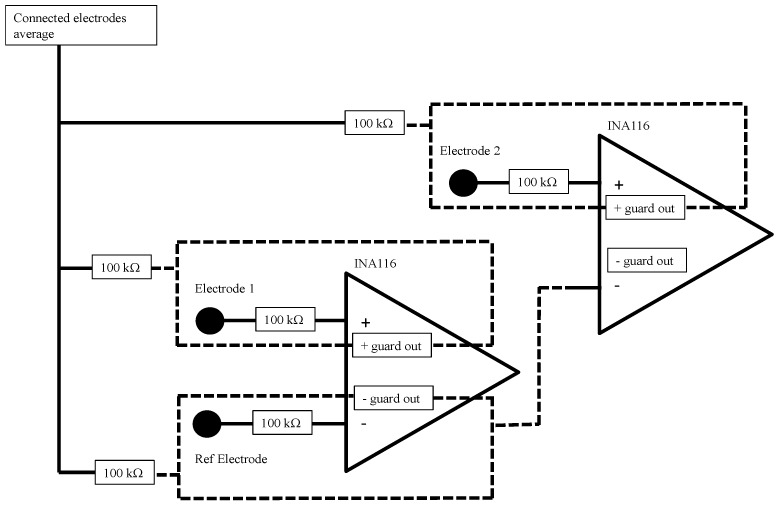
Reference replication, limited to two channels (see text).

**Figure 3 sensors-19-00772-f003:**
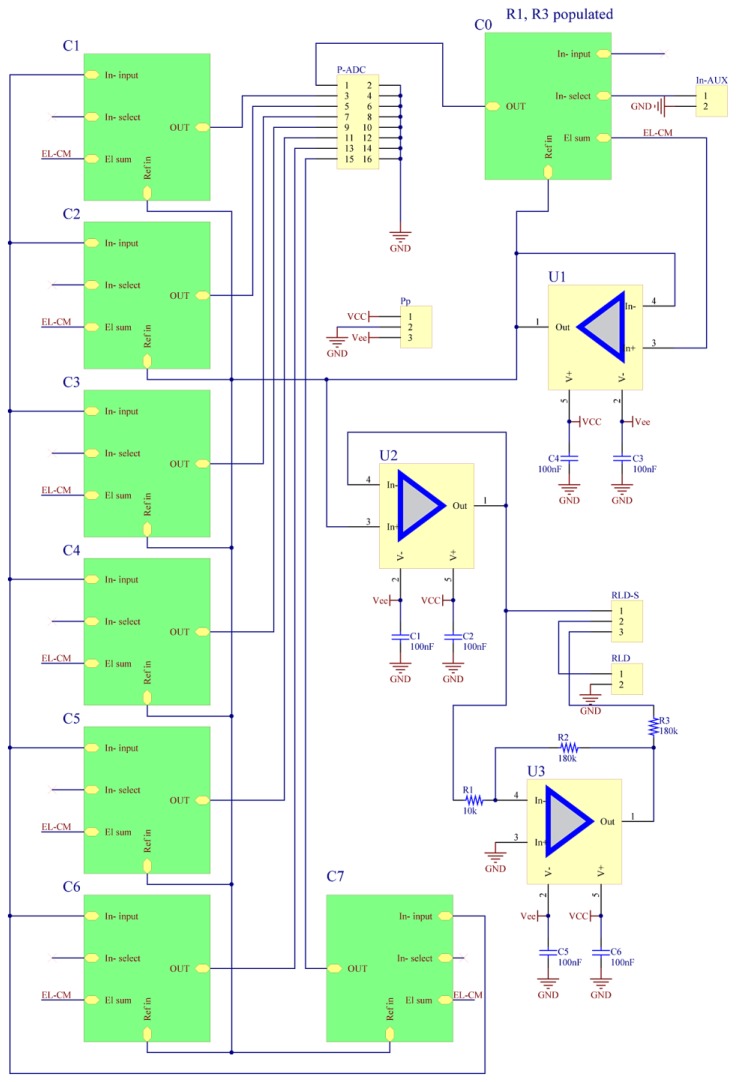
Complete 8-channel design (screen capture from Altium Designer).

**Figure 4 sensors-19-00772-f004:**
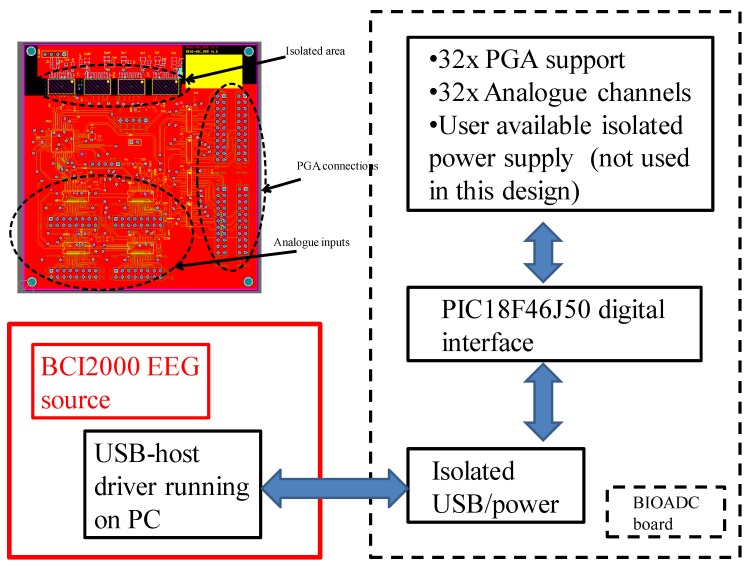
BIOADC main board and block diagram (adapted from [52]). EEG, electroencephalography; CS, chip-select lines; ADC, analogue to digital converter; PGA, programmable gain amplifier.

**Figure 5 sensors-19-00772-f005:**
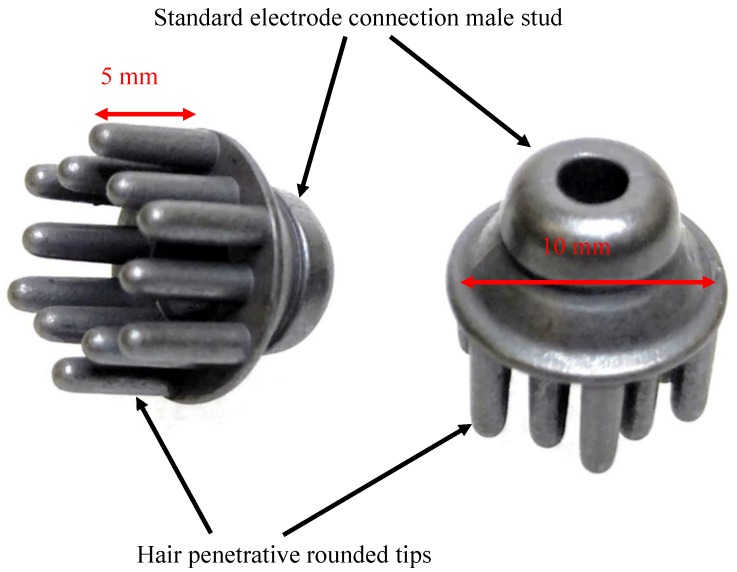
Dry electrodes used for this project.

**Figure 6 sensors-19-00772-f006:**
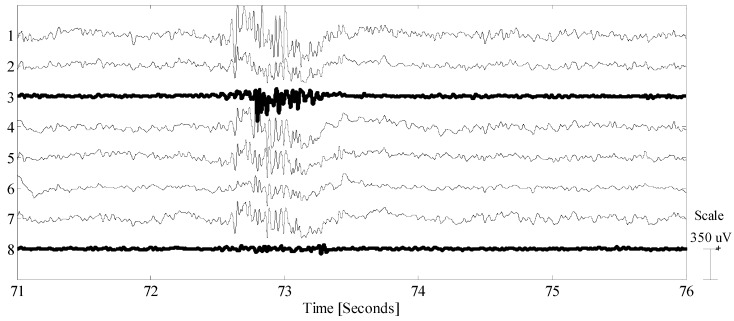
EEG data excerpt including a jaw artifact. Bold traces mark the two electrodes not in good contact. Please note how the scalp-shifts during the artifacts are sufficient to achieve some contact at least for channel 3, which displays a physiological signal only during the artifact.

**Figure 7 sensors-19-00772-f007:**
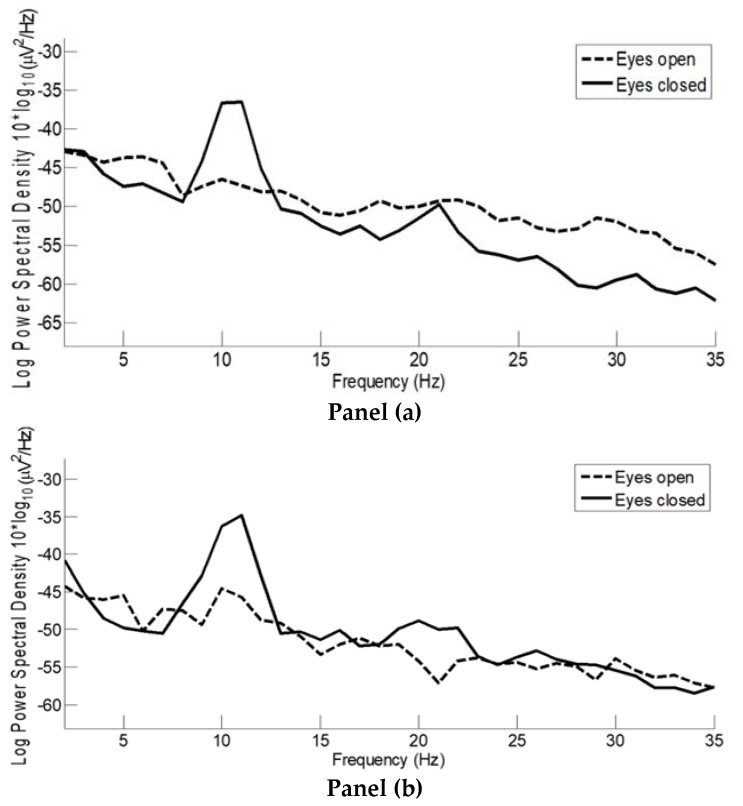
Alpha wave test. Both panels compare 20 s of data selected as artifact-free for the same electrode (very close to the standard 10–20 O1 position) for both eyes open (dashed bold line) and eyes closed (solid bold line). **Panel a**: data recorded using the onboard Driven Right Leg (DRL), power of alpha increase of 1.30 [10log_10_(µV^2^/Hz)/10log_10_(µV^2^/Hz)] times with respect to the baseline; **panel b**: data recorded using direct grounding with the inline protection resistor, power of alpha increase of 1.31 [10log_10_(µV^2^/Hz)/10log_10_(µV^2^/Hz)] times, with respect to the baseline.

**Figure 8 sensors-19-00772-f008:**
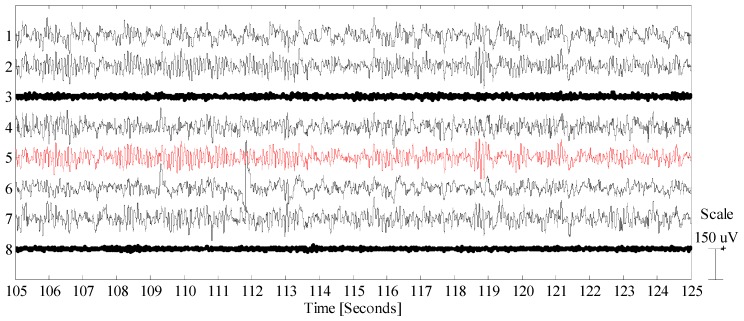
Alpha wave test. In red are the electrodes selected for the spectra analysis. In bold are electrodes not in good contact.

**Figure 9 sensors-19-00772-f009:**
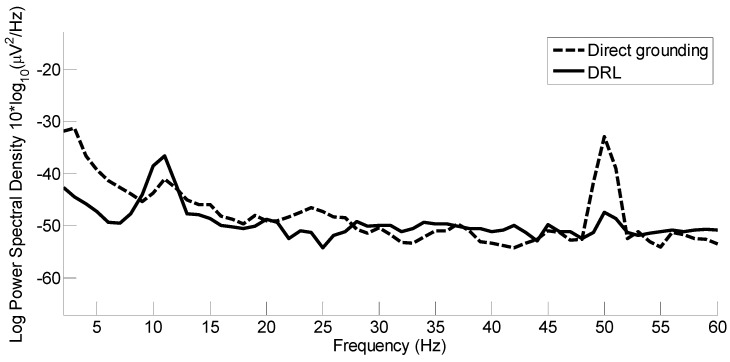
Power-line noise capture comparison. Bold trace is with the use of the Driven Right Leg (DRL), which reduces the power-line noise capture by at least 10 dB.

**Table 1 sensors-19-00772-t001:** Summary of electrical characteristics.

**Number of analogue channels**	8 per module, BIOADC supports 4 modules
**Electrodes compatibility/electrodes configuration**	Passive dry/wet up to contact impedance unbalance of 10^9^ Ω [4,5,6,7,8,9,10,11]/user defined i.e., clustered or 10–20
**Electrode montage**	Differential/unipolar (user selectable)
**Power supply**	Up to ±18 V
**Current consumption**	8.5 mA when powered by ±9 V
**Input impedance**	>10^15^/2 Ω/pF
**Input referred noise (shorted inputs)**	Up to 10 Hz bandwidth 3 µV_pp_
**ADC resolution (BIOADC)**	16-bit (software span-able)

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
