# Peer review of "Fully Open-Access Passive Dry Electrodes BIOADC: Open-Electroencephalography (EEG) Re-Invented"

_sensors, 2019, doi:10.3390/s19040772_

Reviewer 1 Report

This is a well-written draft. The authors clearly describe the hardware details, results and performance. However, the motivation of this study is weak. General speaking, low cost or power consumption, portability (mobility), performance improvement using advanced signal-processing algorithms, and so on are key factors to reinvent a new platform. These factors are all missing in this study. Therefore I don't see the significant contribution in BCI field.

Author Response

Reviewer 1

This is a well-written draft. The authors clearly describe the hardware details, results and performance.

Our answer

Authors wish to acknowledge the reviewer for her/his kind words.

However, the motivation of this study is weak. General speaking, low cost or power consumption, portability (mobility), performance improvement using advanced signal-processing algorithms, and so on are key factors to reinvent a new platform. These factors are all missing in this study. Therefore I don't see the significant contribution in BCI field.

Our answer

With all the due respect, we disagree to some extent about the motivation for this work to be weak. One of the major problems with the current Open-EEG system is the low channels count and the necessity to use the limited number of channels in differential mode. Our system addresses this main issue bringing the channel count to up to 32. Most importantly, our system allows the use of passive dry electrodes that is an enormous advantage in BCI field where the necessity to prepare the head and keep under control contact impedance is the major cause of EEG SNR degradation. We have stressed this concept more in the revised introduction.

We have also added the power consumption measured for our prototype as well as the full electronic characteristics table to the result section as requested. 

Reviewer 2 Report

The idea presented in the paper is rather of engineering nature, but has a high potential to support the progress of science, in particular in experimental BCI interfacing. Making all the design open to the public is also of a great value and may be helpful for many teams with no electronics design support.

The presentation style, however, must be improved before publishing the paper. Some abbreviations and terms must be clarified. Electronic part labels in figures must be enlarged, otherwise it is not possible to follow the presentation of operation. The reader must not be referenced to the text at each figure - the captions should be self-explanatory.  

The content also should be enriched, especially by putting the research in the context of other works in Introduction. The Discussion also could be more developed:
1) authors should provide quantitative results (not only plots) e.g. what was SNR in Alpha range, What was the attenuation of 50Hz power line noise etc.
2) drawbacks of the proposed application should also be mentioned
3) a comparison to existing recording systems would be highly welcome

Please see the list of 23 detailed anonymized comments in attached manuscript.

Author Response

Reviewer 2

The idea presented in the paper is rather of engineering nature, but has a high potential to support the progress of science, in particular in experimental BCI interfacing. Making all the design open to the public is also of a great value and may be helpful for many teams with no electronics design support.

Our answer

Authors wish to acknowledge the reviewer for her/his kind words. We agree that this fully open access hardware will be of interest to the researchers as well as EEG practitioners particularly if electronic support is negligible or absent

The presentation style, however, must be improved before publishing the paper. Some abbreviations and terms must be clarified. Electronic part labels in figures must be enlarged, otherwise it is not possible to follow the presentation of operation. The reader must not be referenced to the text at each figure - the captions should be self-explanatory.  

Our answer

We have improved the figure captions and we have amended the schematic figures to make them more readable as requested. Unfortunately there are limitations with the schematic capture tool. For this reason:

1) We have compacted the schematic diagrams so they are now scaled less (more readable in the limited space allowed on the paper).

2) We have enlarged the native font of the schematic tool where possible.

3) We had already supplied the schematic images as PNG that have no scaling factor, we updated the file.

The content also should be enriched, especially by putting the research in the context of other works in Introduction. The Discussion also could be more developed:

1) authors should provide quantitative results (not only plots) e.g. what was SNR in Alpha range, What was the attenuation of 50Hz power line noise etc.

2) drawbacks of the proposed application should also be mentioned
3) a comparison to existing recording systems would be highly welcome

Our answer

We searched the literature aiming to find fully open access hardware compatible with BCI2000. Aside the referred “Open EEG” we have not been able to find others. We would gladly welcome suggestions to enrich our background/introduction. We have found numerous application of 3-D printable EEG headsets based upon applications of the TI ADS1299 (max 8 channels) and proprietary interfaces that are not yet included in BCI2000. The Input impedance of the ADS1299 range (source Texas Instruments) is inferior to the proposed system (ADS1299 reports 300 pA bias current while INA116 reports femto-Amperes range) hence less suitable for dry electrodes than our system. Open EEG (2 channels only differential mode) is instead compatible with BCI2000 and claims compatibility with active dry electrodes.

Sources:

http://www.ti.com/product/ADS1299

http://openeeg.sourceforge.net/doc/

We have improved the discussion section as requested adding drawbacks of the proposed system particularly in comparison with compact systems based upon the ADS1299. Please note that as these systems are not “open access” we have not included them in our introductory section.

An earlier version of this hardware was compared directly versus a clinical EEG device in a hospital settings with good results. According to https://imotions.com/blog/top-14-eeg-hardware-companies-ranked/ we have already compared a homologous of the proposed system with what is considered to be the best EEG system on the market (Compumedics NeuroScan). Despite our systems employs exclusively dry electrodes, when one of our electrodes is surrounded by other electrodes connected to the predicate device, the correlation between our signal a and the average of the surrounding electrodes is in average of 83% during a BCI task (i.e. Mu-rhythm). As example we have pasted below one of the figures taken from our manuscript (Gargiulo, Gaetano, Rafael A Calvo, Paolo Bifulco, Mario Cesarelli, Craig Jin, Armin Mohamed, and André van Schaik. 2010. 'A new EEG recording system for passive dry electrodes', Clinical Neurophysiology, 121: 686-93.). Of note, what seems to be a relatively small correlation factor is influenced by the slight different area of the scalp “observed” by each individual electrodes as well as the slightly different hardware filters, all these considerations are clearly addressed and stated in (Gargiulo, Gaetano, Rafael A Calvo, Paolo Bifulco, Mario Cesarelli, Craig Jin, Armin Mohamed, and André van Schaik. 2010. 'A new EEG recording system for passive dry electrodes', Clinical Neurophysiology, 121: 686-93.).

In addition, we would like to point out that the human brain is not exactly a signal generator. The power of the alpha waves are influenced by many factors such as the cognitive state. Most importantly, the EEG spectra during eyes open is influenced by even more factors. For this reasons would be extremely difficult to formalize the power and/or SNR for the alpha waves, please see (https://www.ncbi.nlm.nih.gov/pmc/articles/PMC3507158/). However, for the purpose of this review we have quantified the alpha power increase at the max of their spectra (11 Hz in this particular case) measured as ratio of the power of the baseline (eyes open) at the same frequency of 11 Hz divided the measured alpha power.

These measures have been added to the manuscript. For the data in Figure 7a the ratio is 1.30 [10log10(µV2/Hz) / 10log10(µV2/Hz)]; for the data in Figure 7b is 1.31 [10log10(µV2/Hz) / 10log10(µV2/Hz)]

To remove ambiguity in the evaluation of the power-line noise capture we did as suggested referring to the value in dB (see line 276 of the amended manuscript) and also explained better the DRL function in the introduction.

Please see the list of 23 detailed anonymized comments in attached manuscript.

Our answer

We would like to acknowledge the reviewer for her/his efforts. All the comments present in the PDF have been addressed.

Reviewer 3 Report

The paper is well written and clear and it can be of interest to practitioners  in the field.

Author Response

Reviewer 3

The paper is well written and clear and it can be of interest to practitioners  in the field.

Our answer

Authors wish to acknowledge the reviewer for her/his kind words. We agree that this fully open access hardware will be of interest to the researchers as well as EEG practitioners

 Round  2

Reviewer 1 Report

Hi,  Maybe I missed the point again but the value of engineering impact is still weak. It is understandable that multiple channels could benefit some applications. However, it is also known that dry electrodes has poor signal quality. In order to convince readers, could author provide an application with 5+ subjects to demonstrate how this system benefit it? It would be a plus for this system.

In addition, could authors list a table to compare these portable EEG systems? 
1) Cognionics,  ww.cognionics.net

2) Emotive, https://www.emotiv.com/

3) OpenBCI, https://openbci.com/

4) Wearablesensing, http://wearablesensing.com/

5) ABM, https://www.advancedbrainmonitoring.com/

6) mBraintrain, https://mbraintrain.com/ 

with a little engineering effort, it is not too surprise one can build a snap-button-converter so the above mentioned devices can be compatible with vary dry electrodes. I understand these are not open source system (expect OpenBCI), but I believe this is more easily to make authors' system distinguishable. 

Author Response

Once again we would like to acknowledge the reviewer for her/his valuable comments. We added an additional appendix section (starting at line 354) that includes the requested comparison table and example of application. Once again we would like to highlight that one of the major advantages of the proposed system (aside being fully open access) is the record set-up time that is often just the time required to wear the cap.

Reviewer 2 Report

The authors addressed all concerns I raised in the first round. Thanks a lot!

In my opinion the paper improved very much - the motivation has been clearly stated and the results sound much more technically. The prototype, although not yet portable and with moderate energy efficiency is a promissing starting point and therefore worth dissemination. Consequently the paper is expected to attract readers among the developers of BCI interfaces. 

I recommend to publish it after minor revisions of some editorial bugs. No further review is necessary.

Author Response

Authors wish to acknowledge the reviewer kindness and her/his effort that contributed to the improvement of the manuscript.

Round  3

Reviewer 1 Report

This work is now completed to me.